# Exploring standardisation, monitoring and training of medical devices in assisted vaginal birth studies: protocol for a systematic review

Emily J Hotton,[1,2] Sophie Renwick,[2] Katie Barnard,[2] Erik Lenguerrand,[3] Julia Wade,[4] Tim Draycott,[2] Joanna F Crofts,[2] Natalie S Blencowe[5]

[1]Translational Health Sciences, University of Bristol, Bristol, UK
[2]Women's and Children's Research, North Bristol NHS Trust, Bristol, UK
[3]School of Clinical Sciences, Musculoskeletal Research Unit, University of Bristol School of Clinical Science, Bristol, UK
[4]School of Social and Community Medicine, University of Bristol, Bristol, UK
[5]Centre for Surgical Research, School of Social and Community Medicine, University of Bristol, Bristol, UK

**Correspondence to**
Dr Emily J Hotton;
emily.hotton@nhs.net

## ABSTRACT

**Introduction** Assisted vaginal birth (AVB) can markedly improve maternal and neonatal outcomes arising from complications in the second stage of labour. Historically, both forceps and ventouse devices have been used to assist birth; however, they are not without risk and are associated with complications, such as cephalohaematoma, retinal haemorrhage and perineal trauma. As new devices are developed to overcome the limitations of existing techniques, it is necessary to establish their efficacy and effectiveness within randomised controlled trials (RCTs). A major challenge of evaluating complex interventions (ie, invasive procedures/devices used to assist vaginal birth) is ensuring they are delivered as intended. It can be difficult to standardise intervention delivery and monitor fidelity, and account for the varying expertise of clinicians (accoucher expertise). This paper describes the protocol for a systematic review aiming to investigate the reporting of device standardisation, monitoring and training in trials evaluating complex interventions, using AVB as a case study.

**Methods and analysis** Relevant keywords and subject headings will be used to conduct a comprehensive search of MEDLINE, Embase, Cochrane Central Register of Controlled Trials, Cumulative Index of Nursing and Allied Health Literature and ClinicalTrials.gov, for RCTs and pilot/feasibility studies evaluating AVB. Abstracts will be screened and full-text articles of eligible studies reviewed for inclusion. Information relating to the following categories will be extracted: standardisation of device use (ie, descriptions of operative steps, including mandatory/flexible parameters), monitoring of intervention delivery (ie, intervention fidelity, confirming that an intervention is delivered as intended) and accoucher expertise (ie, entry criteria for participation, training programmes and previous experience with the device). Risk of bias of included studies will be assessed.

**Ethics and dissemination** Ethical approval is not required because primary data will not be collected. Findings will be disseminated by publishing in a peer-reviewed journal and presentations at relevant conferences.

## Strengths and limitations of this study

► This review will include all randomised controlled trials (RCTs)/feasibility studies evaluating assisted vaginal births (AVBs), regardless of the nature of the comparator, ensuring that all AVB data are captured.
► Specifically, the review will summarise reporting standards relating to standardisation and monitoring of intervention delivery, and ways in which trials describe and account for clinician expertise in RCTs involving devices.
► The review is not limited to human studies, ensuring that any relevant AVB study is included.
► No language limitations have been set, ensuring that the review is as comprehensive and generalisable as possible.
► This review focuses only on RCTs and pilot/feasibility studies, meaning that information from other study designs may be missed.

## INTRODUCTION

Assisted vaginal birth (AVB) is a vital procedure that, in skilled hands, can markedly reduce maternal and neonatal complications in the second stage of labour.[1] In the UK, approximately one in eight women require an AVB, which typically involves forceps and/or ventouse devices.[2] However, AVB is not without risk. A forceps-assisted birth confers an increased risk of perineal and vaginal trauma[3 4] as well as faecal incontinence.[4 5] Ventouse-assisted births have a failure rate of approximately 30% as well as being associated with neonatal subgaleal haematoma and intracranial haemorrhage, leading to a statutory warning in 2015 by the Food and Drug Administration.[4] These problems, together with the threat of litigation, have contributed to a reduction in AVB rates worldwide. There has been a corresponding increase in Caesarean section rates, despite the fact that AVB often provides better outcomes at full dilation and prevents future problems, such as increased risk of abnormal placentation, scar rupture and unexplained stillbirth in subsequent pregnancies.[6 7] Novel AVB devices may be able to address these known risks and

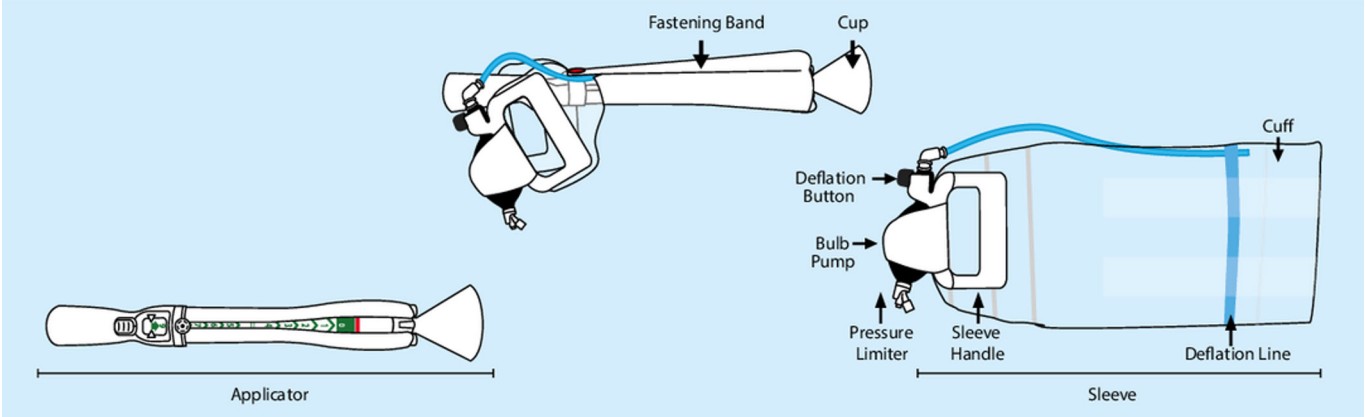

**Figure 1** BD Odon Device components.

attempt to transform the falling AVB rates worldwide. One example is the BD Odon Device. The device has an air cuff that, once placed around the baby's head, is inflated. To assist the birth of the baby, the accoucher then applies traction on the sleeve, which is attached to the air cuff (figure 1). In contrast to the ventouse, which operates by exerting negative pressure on the baby's head, the BD Odon Device exerts positive pressure via the air cuff. It is hypothesised that this may reduce neonatal intracranial bleeding, and that the circumferential positioning of the air cuff may reduce instrumental failure rates.

Despite the perceived benefits of novel devices, such as the BD Odon Device, novel devices are susceptible to 'optimism bias'. Optimism bias refers to the unjustified belief in 'new or novel' innovations.[8] It is, therefore, necessary for all pioneering technologies to undergo rigorous evaluation to ensure that the benefits and harms are fully investigated and establish whether they are better than the standard devices used in clinical practice. Many expert panels, including the European Clinical Research Infrastructure Network, have suggested that more rigorous clinical evaluation of medical devices within randomised controlled trials (RCTs) is required.[9–12] Currently, however, the pathway for evaluating novel procedures and devices is less distinct than that for pharmaceutical products, and specific barriers have been identified in undertaking RCTs in this area.[13] A major challenge is that they are considered to be complex interventions—defined as those with multiple interacting components that can act independently or interdependently to influence outcomes. This can create difficulties in establishing how the intervention should be delivered (standardisation) and ascertaining whether it is actually delivered as intended (intervention fidelity). An additional challenge is that the delivery of complex interventions can be influenced by clinicians' skill.

These issues have been acknowledged in reporting guidance documents, such as the CONSORT extension for non-pharmacological treatments (CONSORT-NPT).[14] CONSORT-NPT suggests that 'precise details of the experimental treatment', 'details on whether and how the interventions were standardised', 'eligibility criteria

for care providers', 'the number of care providers', 'a description of care providers expertise and qualification' and 'the number of patients treated by each care provider' are reported.[14] Additionally, 'details of whether and how adherence of care providers to the protocol and of participants to interventions was assessed' is recommended.[14] Provision of this information is recommended to improve the quality of trial design and to enable successful interventions to be replicated in practice, improving the contextualisation of findings and reducing research waste. Currently, however, it is uncertain as to whether these reporting standards are met in RCTs involving complex interventions, such as devices. This study, therefore, aims to investigate the quality of reporting of intervention standardisation, monitoring and clinician expertise in trials involving devices, using AVB as a case study.

## METHODS AND ANALYSIS
The review will be conducted in line with the Preferred Reporting Items for Systematic Reviews and Meta-Analyses checklist.[15]

### Eligibility criteria
Feasibility studies, pilot studies and RCTs will be included in the review if they meet the following inclusion criteria:

### Participants
All females of any age having an AVB. Studies involving simulated patients or animals will also be included.

### Intervention
AVB by forceps, vacuum extraction or a novel assisted birth device. All devices will be considered and will not be limited to a single type or manufacturer.

### Comparator(s)
Comparator groups will include spontaneous vaginal birth, AVB using any device or Caesarean section. Pilot/feasibility studies without a comparator group will also be included.

## Outcome(s)

Reporting standards relating to standardisation of device use, monitoring of whether the device was used as intended (intervention fidelity) and details of accoucher expertise will be extracted. Information about the 'success' and 'failure' rates of the device, and adverse events, will also be collected.

### Search strategy and study selection

We will systematically search for RCTs involving AVB device(s) in MEDLINE, Embase, Cochrane Central Register of Controlled Trials, Cumulative Index of Nursing and Allied Health Literature and ClinicalTrials. gov databases from inception to 30 November 2018. The computer-based searches will combine free text and subject headings (see online supplementary file).

Reference lists of included studies will be searched for additional relevant articles, including published protocols. There will be no restrictions on language.

### Identification and selection of papers

A customised inclusion/exclusion form will be used to screen abstracts and provide an audit trail. Titles and abstracts will be screened independently by two authors (EJH and NSB). Any conflicts will be resolved by discussion.

The full-text versions of papers retained after title and abstract screening will be screened for further assessment of their eligibility for inclusion.

### Data extraction and management

Data will be extracted independently by at least two assessors for each paper (EJH, SR and NSB). A customised data extraction form will be used to collect relevant data from each paper. Data of interest will include general study details (author, year of publication and country of origin of study), details of study design (RCT, pilot or feasibility study), the number of participating centres and the total number of participants.

### Standardisation of intervention delivery

Details of the device(s) and comparator(s) will be extracted. These will include verbatim descriptions relating to how the device should be delivered (including technical or operative steps) and how/whether this was standardised within the study. Details concerning the criteria for using the device, such as any mandatory, prohibited or flexible parameters, will be documented in accordance with an existing typology for considering standardisation of interventional procedures.[16] Finally, assessors will record judgements about whether enough information is provided to be able to replicate device use in routine practice (yes/no/unsure).

### Monitoring of whether the device was used as intended (intervention fidelity)

Any reporting of whether the device was used as intended (intervention fidelity) will be reported. Details of how intervention fidelity was measured will be documented (eg, within case report forms).

### Accoucher expertise

The number of accouchers participating in the study, and delivering interventions in each trial group, will be recorded. If provided, the total number of births (and AVBs) in each study centre will be reported. Reporting of any information about accoucher expertise will be recorded including their grade, previous experience with the device(s) under investigation and any protocols for supervision when using the device. Attempts to account for a potential learning curve in device delivery (eg, trial entry criteria for accouchers, such as a prespecified number of deliveries) will be recorded, together with information about accoucher training (eg, mandatory courses, videos or other materials). Finally, accoucher-related outcomes, such as competence, confidence or knowledge, will be extracted.

### Device success, failure and safety

Details of whether the device was used successfully will be recorded, together with information about 'harms' or 'adverse events' in either women or their babies. Information about causes or reasons for these events will be extracted verbatim.

### Assessment of study quality

The Cochrane risk of bias tool will be used to evaluate bias in RCTs, and pilot or feasibility studies that involved randomisation.[17] Non-randomised pilot and feasibility studies will be assessed by evaluating bias related to the process of trial recruitment, documentation of protocol non-adherence, reporting of a primary outcome, description of clear objectives and description of clear progression criteria.

### Data synthesis and statistical analysis

Data will be entered into a custom database. A narrative synthesis will summarise the findings. Any further data synthesis (such as meta-analyses) will depend on the number and quality of studies identified.

### Patient and public involvement

Patients and the public were not involved in the design and development of this protocol.

## ETHICS AND DISSEMINATION

The completed systematic review will be published in a peer-reviewed journal and presented at appropriate conferences. This protocol can further be adapted for the analysis of other devices within obstetrics and surgery.

This systematic review will provide important information surrounding the quality of reporting in RCTs evaluating devices for AVB, relating to how device use is standardised in trials (standardisation), whether devices are used in trials as intended (monitoring/intervention fidelity) and what the level of accoucher training is. The

findings will inform the design of future pilot/feasibility studies and/or RCTs in this area, by optimising the way that device use is standardised and monitored, and accoucher expertise is accounted for.

**Contributors** EJH and NSB initiated and designed the study with methodology inputs from EL, JW, TD and JFC. EJH and SR performed the data collection. KB performed database searches. EJH and NSB drafted the manuscript with inputs from SR, KB, EL, JW and JFC. All authors contributed to revisions of the manuscript and approved the final version.

**Funding** This work was supported by the Bill & Melinda Gates Foundation [grant number OPP1184825]. EJH, SR, JFC and TD are employees of the North Bristol NHS Trust, which receives funding from PROMPT Maternity Foundation (PMF) to pay part of their salaries. PMF has received funds from BD, manufacturer of the Odon device. EL is an employee of the University of Bristol, which receives funding from PMF to pay part of EL's salary. NSB is an NIHR Clinical Lecturer. This study is being supported by the NIHR Biomedical Research Centre at the University Hospitals Bristol NHS Foundation Trust and the University of Bristol, and the MRC ConDuCT-II (Collaboration and innovation for Difficult and Complex randomised controlled Trials In Invasive procedures) Hub for Trials Methodology Research (MR/K025643/1).

**Competing interests** None declared.

**Patient consent for publication** Not required.

**Provenance and peer review** Not commissioned; externally peer reviewed.

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
