## [Reviewer comments · BMJ Open]

ARTICLE DETAILS

TITLE (PROVISIONAL)	Exploring standardisation, monitoring and training of medical devices in assisted vaginal birth studies: protocol for a systematic review
AUTHORS	Hotton, Emily; Renwick, Sophie; Barnard, Katie; Lenguerrand, Erik; Wade, Julia; Draycott, Tim; Crofts, Joanna; Blencowe, Natalie

VERSION 1 - REVIEW

REVIEWER	Michel Boulvain Hôpitaux Universitaires de Genève, Switzerland I am involved in the process of developing the Odon device. I collaborate with the team in Bristol, and co-authored some papers describing the process of evaluation of the device on mannequins. I am not involved, however, in the present protocol. I have no Financial ties with BD (the company developing the Odon device), other than reimbursement of travel to meetings. I have no Financial ties with Bristol.
REVIEW RETURNED	11-Dec-2018

GENERAL COMMENTS	This systematic review is welcome, to inform the development and evaluation of the effectiveness and potential harms of any newly developed instrument to assist the delivery process. I would be curious to see the original papers describing the development of forceps or the original vacuum extractors. I have no specific comments to make on the protocol of this systematic review. Please see above my COI declaration and I let you judge on whether it is appropriate or not that I comment on this protocol.
---

REVIEWER	Mónica García-Sevilla Universidad Carlos III de Madrid, Spain
REVIEW RETURNED	19-Feb-2019

GENERAL COMMENTS	Interesting proposal to review the current use of devices for AVB and the outcomes of the most common devices as well as a new one. Everything was correctly presented. Just missing some dates of the study.
---

REVIEWER	Jerome Cornette O&G Erasmus MC, Rotterdam
REVIEW RETURNED	22-Feb-2019

GENERAL COMMENTS	I read the protocol with pleasure. This protocol describes the intention for a systematic review according to PRISMA Checklist. It will search for various types of studies (randomised controlled trials and pilot/feasibility studies) on different types of assisted deliveries (vacuum extraction, forceps deliveries, alternatives (eg ODon device) and asses IF and HOW standardization of device use (i.e. descriptions of operative steps, including mandatory/flexible parameters); monitoring of intervention delivery (i.e. Intervention fidelity, confirming that an intervention is delivered as intended), and accoucher expertise (i.e. entry criteria for participation, training programmes, previous experience with the device)as well as outcomes and complications are reported. This knowledge can then help in defining optimal standards on these issues for future trial desing. The aims of this study are certainly very relevant. Assisted deliveries are performed by the thousands on a daily base. Nevertheless knowledge on their use, role, type, benefits and associated risks are often more colored by perceptions, both amongst professionals and lays, rather than based on scientific evidence. The complex interplay between the above mentioned factors (indications, monitoring, type (high , low rotational,...) accoucher expertise) along with societies ever changing obstetric standards probably contribute to this. It is certainly recommendable to learn from previous work as to offer recommendations and improve future research desing for studies comparing existing and new devices for assisted delivery. The protocol is well written and will proceed (methodology) according to scientific internationally accepted standards for systematic review.(Prisma). I especially applaud the fact that there will be no restriction to language in the manuscript selection as national customs can be very different on the issue of assisted deliveries, which is nevertheless performed by the thousands a day over the world.
--

VERSION 1 – AUTHOR RESPONSE

Reviewer 1 comments

This systematic review is welcome, to inform the development and evaluation of the effectiveness and potential harms of any newly developed instrument to assist the delivery process. I would be curious to see the original papers describing the development of forceps or the original vacuum extractors.

I have no specific comments to make on the protocol of this systematic review.

Reply: We thank the reviewer for their support.

Reviewer 2 comments

Interesting proposal to review the current use of devices for AVB and the outcomes of the most common devices as well as a new one. Everything was correctly presented. Just missing some dates of the study.

Reply: We thank the reviewer for this comment and have now clarified this in the text

Revisions: We have added the specific dates for searching. The sentence now reads:

“We will systematically search for RCTs involving AVB device(s) in Medline, EMBASE, Cochrane Central Register of Controlled Trials (CENTRAL), Cumulative Index of Nursing and Allied Health Literature (CINAHL) and ClinicalTrials.gov databases from inception to 30th November 2018.”

Reviewer 3 comments

I read the protocol with pleasure.

This protocol describes the intention for a systematic review according to PRISMA Checklist. It will search for various types of studies (randomised controlled trials and pilot/feasibility studies) on different types of assisted deliveries (vacuum extraction, forceps deliveries, alternatives (e.g. ODon device) and assess IF and HOW standardization of device use (i.e. descriptions of operative steps, including mandatory/flexible parameters); monitoring of intervention delivery (i.e. Intervention fidelity, confirming that an intervention is delivered as intended), and accoucher expertise (i.e. entry criteria for participation, training programmes,

previous experience with the device) as well as outcomes and complications are reported. This knowledge can then help in defining optimal standards on these issues for future trial design. The aims of this study are certainly very relevant. Assisted deliveries are performed by the thousands on a daily basis. Nevertheless knowledge on their use, role, type, benefits and associated risks are often more coloured by perceptions, both amongst professionals and lay, rather than based on scientific evidence. The complex interplay between the above mentioned factors (indications, monitoring, type (high, low rotational,...) accoucher expertise) along with societies ever changing obstetric standards probably contribute to this. It is certainly recommendable to learn from previous work as to offer recommendations and improve future research design for studies comparing existing and new devices for assisted delivery. The protocol is well written and will proceed (methodology) according to scientific internationally accepted standards for systematic review (Prisma). I especially applaud the fact that there will be no restriction to language in the manuscript selection as national customs can be very different on the issue of assisted deliveries, which is nevertheless performed by the thousands a day over the world.

Reply: We thank the reviewer for this kind comment.